# Progressive Respiratory Insufficiency in a Teenager with Diaphragmatic Hypomotility Due to a Novel Combination of Gliomedin Gene Variants

**DOI:** 10.3390/children9060797

**Published:** 2022-05-28

**Authors:** Benjamin Eurich, Catharina Nitsche, Margot Lau, Britta Hanker, Juliane Spiegler, Guido Stichtenoth

**Affiliations:** 1Department of Pediatrics, University of Lübeck, Ratzeburger Allee 160, 23538 Lübeck, Germany; catharina.nitsche@uksh.de (C.N.); margot.lau@uksh.de (M.L.); guido.stichtenoth@uksh.de (G.S.); 2Division of Pediatric Pneumology & Allergology, University Hospital Schleswig-Holstein Campus Lübeck, Airway Research Center North (ARCN), German Center for Lung Research (DZL), Ratzeburger Allee 160, 23538 Lübeck, Germany; 3Institute of Human Genetics, University of Lübeck, Ratzeburger Allee 160, 23538 Lübeck, Germany; britta.hanker@uksh.de; 4Department of Pediatrics, Julius Maximilian University of Würzburg, Sanderring 2, 97070 Würzburg, Germany; spiegler_j@ukw.de

**Keywords:** GLDN variant, gliomedin, juvenile progressive respiratory insufficiency, diaphragmatic hypomotility, scoliosis, arthrogryposis, LCCS11, axonopathy, FADS

## Abstract

Lethal congenital contracture syndrome 11 (LCCS11) is a form of arthrogryposis multiplex congenita (AMC) which is associated with mutations in the gliomedin gene (GLDN) and has been known to be severely life-shortening, mainly due to respiratory insufficiency. Patients with this condition have been predominantly treated by pediatricians as they usually do not survive beyond childhood. In this case report, we present a young adult who developed severe progressive respiratory insufficiency as a teenager due to diaphragmatic hypomotility and was diagnosed with LCCS11 following the discovery of compound heterozygous pathogenic variants in GLDN. This case demonstrates the importance of screening for neuromuscular diseases in well-child visits and follow-ups of patients at risk for gross and fine motor function developmental delay. It also underscores the significance of including LCCS11 and other axonopathies in the differential diagnosis of juvenile onset of respiratory insufficiency, highlights that patients with this condition may present to adult practitioners and questions whether the nomenclature of this condition with various phenotypes should be reconsidered due to the stigmatizing term ‘lethal’.

## 1. Introduction

Development of the peripheral nervous system (PNS) is promoted by gliomedin, a transmembrane and secreted protein encoded by the gliomedin gene (GLDN), which facilitates the formation of the nodes of Ranvier by promoting clustering of sodium channels [1,2]. Biallelic mutations in this gene were first associated with a putatively lethal form of arthrogryposis multiplex congenita (AMC), lethal congenital contracture syndrome 11 (LCCS11), presumably due to pulmonary hypoplasia attributable to reduced fetal mobility because of defects in the development of the PNS [3].

In 2017, it was shown that the range of LCCS11 phenotypes includes patients with long-term survival beyond childhood receiving long-term respiratory and nutritional assistance [4].

In this case report, we present a patient with a novel combination of GLDN variants who developed progressive respiratory insufficiency during adolescence due to increasing diaphragmatic hypomotility and severe scoliosis.

We also highlight the importance of well-child visits and regular follow-ups as a crucial part of interdisciplinary prevention measures installed by a healthcare system, especially for patients at risk of gross and fine motor function developmental delay. Thus, neuromuscular diseases can be diagnosed early on, allowing multi-disciplinary disease management, including therapy [5].

## 2. Case Report

### 2.1. Initial Presentation at the Emergency Department

The patient first presented to our emergency department at the age of 16 years after a syncopal event at school. Physical examination showed a cachectic male teenager with signs of acute and chronic hypoxia, including tachydyspnea, cyanosis of the lips and clubbed fingers. The patient’s history revealed rapidly decreasing physical ability and increasing respiratory insufficiency.

The presentation in late adolescence with such severe, chronic and progressing signs and symptoms was unusual, and led to a detailed diagnostic process following admission to the intermediate care ward.

### 2.2. Medical, Family and Psychosocial History, including Relevant Genetic Information

The patient was born prematurely at 33 weeks’ gestation due to preterm premature rupture of membranes and oligohydramnios. He showed signs of muscular hypotonia, including dysphagia, and arthrogryposis, in particular contractures of hip, knee, elbow and hand joints, as well as a unilaterally dislocated hip joint. Chest deformity and cryptorchidism were also found. The birth weight was appropriate for the gestational age. Postnatally, non-invasive ventilation using continuous positive airway pressure (CPAP) was required for 5 weeks.

Extensive investigations conducted at the time into the possible etiology of the signs and symptoms, including MRI of the brain, EEG and laboratory testing for metabolic diseases, revealed no relevant pathology. Genetic analyses showed neither karyotypic abnormalities nor a homozygous loss in the SMN1 gene found in spinal muscular atrophy (SMA).

Follow-ups until 2 years of age were conducted by a general practitioner. At the age of one year, developmental delay was recognized in speech, motor development and growth. The milestone “walking without support” was not achieved until the age of two; thus, physiotherapy and participation in the Early Support for Infants and Toddlers program were initiated. Motor competence improved continuously until the age of 16; however, muscle fatigue increased with age and strenuous physical activity was mainly avoided.

To correct thoracolumbar kyphoscoliosis, which had previously been treated with a thoracolumbosacral orthosis since the age of four years, spondylodesis was performed at the age of 10 years—without further monitoring of neuromuscular development or interdisciplinary management.

### 2.3. Clinical Findings

Pulmonary function testing (PFT) at the age of 16 years yielded a restrictive pattern and hyperinflated lungs with a reduced overall volume, as well as ventilation inhomogeneity (represented by an elevated lung-clearance index). Bronchoscopy and MRI did not reveal any obstruction of the large airways, nor did it display compression of the large thoracic vessels nor any abnormality of the lung tissue, but clinically relevant progressive proximal junctional kyphosis (PJK) and global muscle hypotrophy putatively promoting scoliosis were shown. Sonography disclosed diaphragmatic hypomotility.

Nocturnal non-invasive home ventilation had been commenced already due to hypopnea at night.

We concluded that the increasing respiratory insufficiency and the PTF results were mainly caused by diaphragmatic hypomotility and severe scoliosis, and that the nocturnal hypopnea contributed to signs of chronic hypoxia.

Neuromuscular examination showed indications of axonopathy: proximal muscular weakness; fasciculation and shivering; contractures of the hands, calves and hamstrings; and paresthesia of the fingers as well as erythematous hands, likely representing loss of autonomic function of the vessels (Figure 1).

Comorbidities included growth restriction, delayed puberty, enuresis, adjustment disorder, depression, eating disorder, gynecomastia and hiatal hernia. Although not formally assessed, it was assumed that the limited autonomy caused reduced quality of life.

Laboratory testing showed elevated CK, LDH and myoglobin. No abnormalities explaining the origin of the symptoms were found in a muscle biopsy. Differential diagnoses, including cystic fibrosis and spinal muscular atrophy, were ruled out.

Extensive genetic analyses were performed, with exome sequencing ultimately revealing a novel combination of GLDN variants, which was highly likely to be the cause of the symptoms, confirming the suspected diagnosis of axonopathy.

The study showed two heterozygous variants NM_181789.4: c.82G > C, *p*. (Ala28Pro) and NM_181789.4: c.1178G > A, *p*. (Arg393Lys). The mother is a heterozygous carrier of the variant c.1178G > A, *p*. (Arg393Lys).

The first variant is estimated to be present in 0.0017% of the general population [6]. Five individuals from two Canadian Inuit families showing an LCCS11 phenotype have been reported with a homozygous genotype for this missense variant. Four of them died shortly after birth, while one survived, with no chronic medical issues at the age of 10 years [7].

The latter variant has a minor allele frequency of 0.0036%. It affects a canonical splicing site at the 5’ end of the 9th out of 10 exons of the gliomedin gene [8]. Thus, exon skipping is likely to occur and, subsequently, a frameshift, premature termination of translation and nonsense-mediated mRNA decay. The variant has been described in association with LCCS11 in a patient by Wambach et al. [4], Mis et al. [9] and Pergande et al. [10], each in conjunction with another likely pathogenic GLDN variant (NM_181789.4: c.1428C > A, *p*. (Phe476Leu) and NM_181789.4: c.1093C > T, *p*. (Leu365Phe), respectively). The patient described by Wambach et al. died at 12 h from respiratory failure due to pulmonary hypoplasia. The one described by Mis et al. also displayed lung hypoplasia with consecutive respiratory failure but could be weaned off respiratory support by 14 months and was alive at 44 months at the time of publication. The patient presented by Pergande et al. with the same compound heterozygosity as the one reported by Mis et al. had no pulmonary hypoplasia reported but multiple joint contractures, delayed motor development, muscular hypertonia and dysphagia, and was alive at 1 year at the time of publication [4,9,10].

### 2.4. Medical Course

Without any causal treatment available, therapy focused on managing symptoms.

A stumbling gait had been attributed to scoliosis for many years before proximal muscle weakness became apparent in late adolescence; at the age of 17 years the patient was wheelchair dependent. Transferring to and from the wheelchair was possible until the age of 18 years. To maintain some independence in terms of mobility, an application for an electric wheelchair has been submitted; approval of the health-insurance company is still pending.

The patient’s non-invasive home ventilation was adjusted, introducing a mouthpiece for a more convenient ventilation in dyspneic phases during the day, implementing a volume guarantee for safety reasons as well as increasing the peak inspiratory pressure of the BiPAP ventilation mode to reduce the work of breathing.

High-calorie components were added to the diet to counterbalance the energy expenditure. Medication with desmopressin and propiverin to treat enuresis and with melatonin to prevent night-time anxiety leading to increased dyspnea were continued.

Spinal surgery was not an option because of the rigidity of the thorax and the ensuing perioperative risks. Diaphragm pacing was discussed but was not an option at the time because of lack of evidence in pediatric neuromuscular diseases.

Psychotherapy was strongly recommended. Unfortunately, the COVID-19 pandemic with the strict contact restriction exercised by the family has led to an isolation and prevented effective psychological support.

The patient showed reduced activity and participation in relevant domains identified by the World Health Organization, e.g., mobility, self-care, education, community and social life, as well as interpersonal interaction [11].

After discharge from hospital, the patient was monitored closely, including regular visits in our pulmonary clinic. Despite optimizing the ventilation settings, the patient developed increasing dyspnea due to progressive diaphragmatic hypomotility and amyosthenia.

### 2.5. Current Status

At the time of publication, the patient is alive at 19 years and requires non-invasive ventilation at night and increasingly during the day, as well as higher levels of oxygen supplementation because of respiratory insufficiency. He has received a mechanical insufflation/exsufflation (MI-E) device to assist with coughing and therefore to support mucus clearance and prevent respiratory tract infections. The patient’s level of autonomy is gradually decreasing due to reduced physical abilities and resulting fatigue. Therefore, attending school regularly and participation in classes has been challenging. To graduate from high school, a sheltered workshop instead of regular classes is being evaluated.

Following the patient’s 18th birthday, transition from pediatric to adult medicine occurred. Gross motor function rapidly decreased, including ability to walk. Thus, at the time of publication he depends on a wheelchair and needs nasal ventilation during daytime. Interestingly, the patient reports intermittent improvement of paresthesia due to vibroacoustic stimulation.

### 2.6. Prognosis

Due to the lack of data on the novel combination of GLDN variants and patients with LCCS11 who survived beyond adolescence, establishing a prognosis is challenging.

Considering the patient’s progressively deteriorating general (and especially respiratory) state of health, the need for optimized support measures as well as psychosocial support is evident. If the need for ventilation throughout the day were to continue, tracheotomy would be an option to allow for enhanced patient comfort during continuous ventilation.

## 3. Discussion

We present a patient with compound heterozygous GLDN variants c.82G > C, *p*. (Ala28Pro) and c.1178G > A, *p*. (Arg393Lys) who was diagnosed with LCCS11 in his adolescence due to progressive respiratory insufficiency because of increasing diaphragmatic hypomotility.

The patient was born with an AMC phenotype; however, an extensive genetic work-up in 2003 did not lead to a diagnosis. At the time of the patient’s birth, lethal arthrogryposis due to neuromuscular disorders had been described [12], but AMC in conjunction with chronic neuromuscular disease had not been reported. Lethal forms of AMC with pulmonary hypoplasia, facial abnormalities, camptodactyly, polyhydramnios and intrauterine growth restriction (IUGR) were summarized under the umbrella of Pena–Shokeir syndrome type 1 or fetal akinesia deformation sequence (FADS), but the various etiologies of this phenotype remained unclear [13,14]. Postnatally, FADS was not considered a possible diagnosis for this patient, presumably because of survival beyond the neonatal period, lack of polyhydramnios and IUGR, as well as successful weaning off non-invasive ventilation, even though the respiratory support was prolonged for the degree of prematurity.

Since extensive investigations did not yield a diagnosis and the symptoms were improving, no further neurological follow-up occurred. Nevertheless, the patient showed fine and gross motor function delay and delayed speech development in well-child visits. He received physiotherapy and took part in the early intervention program, but at this time, a referral to a pediatric neurologist for further diagnostics was missed. Interdisciplinary collaboration and subsequent further diagnostics were also omitted when spondylodesis was performed at the age of 10 years and treatment for nocturnal enuresis commenced during childhood.

In this case, the unusual medical course with late progression of respiratory symptoms caused by increasing diaphragmatic hypomotility led to a reassessment during late adolescence. By then, due to advances in genetic diagnostics and therefore understanding of pathology, adequate testing methods such as whole-exome sequencing, which can identify the cause of developmental delay of previously unknown etiology in more than 40% of children [15], were available. Knowledge of association of the presented symptoms with GLDN variants had also emerged in the meantime, eventually yielding the diagnosis of LCCS11. So far, only one other patient showing similar progressive clinical symptoms during adolescence has been described (Figure 2); however, in that case, diaphragmatic paralysis had already been observed within the first three months [4].

This presentation highlights the importance of including axonopathies such as LCCS11 in the differential diagnosis of neuromuscular diseases in general, but also of respiratory insufficiency—even if the patient presents during adolescence or early adulthood. The exact pathophysiology of respiratory insufficiency in this disease has not been assessed formally, but reduced muscle stimulation due to axonopathy, leading to muscular atrophy and weakness, inclusive of intercostal muscles and the diaphragm, can be assumed a possible mechanistic explanation. Resulting diaphragmatic hypomotility may initially present as hypoventilation solely at night, since diaphragmatic function is crucial for efficient ventilation during sleep [16]. Therefore, assessment of diaphragmatic function via sonography (to evaluate diaphragmatic motility and thickness) and/or twitch mouth (non-invasive) or transdiaphragmatic (invasive) pressure following cervical magnetic stimulation of the phrenic nerves, as well as screening for nocturnal hypoventilation and hypoxemia, should be considered as part of regular follow-ups in addition to pulmonary function testing in patients with signs of respiratory insufficiency [17,18,19]. Hiatal hernia, as seen in this patient, could be secondary to muscle wasting caused by diaphragmatic hypomotility, and therefore is a warning sign. Additionally, chest wall deformities, induced by limited fetal movement and exacerbated by wasting of trunk muscles, can aggravate restricted chest-wall expansion and thereby inefficient inspiration as well as forced expiration, which can lead to insufficient clearing of mucus, resulting in respiratory tract infections and dyspnea. Since patients with neuromuscular disease, especially when accompanied by chest-wall deformity, are at risk of respiratory exacerbation, respiratory function should be monitored frequently to allow for early therapy adjustment.

Treatment options for respiratory insufficiency in LCCS11 (and neuromuscular disease in general) comprise ventilation, cough assistance, physiotherapy, and potentially diaphragm pacing. In early stages of the disease, when nocturnal hypoventilation is predominant, non-invasive ventilation may be sufficient. Using BiPAP rather than CPAP ventilation can reduce the work of breathing. If nocturnal hypoxemia is detected, oxygen supplementation can improve sleep quality. In later stages, when ventilation is required during the daytime and almost continuously due to progressive diaphragmatic hypomotility and impaired chest-wall expansion, non-invasive ventilation may interfere with communication and food intake, especially when ventilation through nasal prongs is insufficient and a mask is required. Tracheostomy and subsequent invasive ventilation can alleviate these issues and reduce anatomic dead space. Cough-assisting appliances such as mechanical insufflation/exsufflation devices as well as manual techniques help with forced expiration and promote mucus clearance, hence preventing respiratory tract infections and aggravated dyspnea. Additionally, physiotherapy—specifically respiratory muscle training using pressure threshold breathing devices—has been shown to be effective in neuromuscular disease [20]. Diaphragm pacing following electrode implantation may potentially have a beneficial effect in patients with neuromuscular disease; however, diaphragm pacing has so far not proven to be neither effective nor safe in studies mainly conducted in ALS patients [21].

Timely diagnosis would not have led to a curative therapy in our patient. However, participation could have been improved by an interdisciplinary and multi-professional follow-up. Timely initiation of respiratory, physical and occupational therapy, adequate caloric intake and optimized assisting devices would probably have improved the participation domains of mobility and self-care. Participation in education, community and social life, as well as interpersonal interaction, was not only impaired by LCCS11, but further hindered by the COVID-19 pandemic.

Due to the late diagnosis, the patient was confronted with uncertainty regarding the etiology of his symptoms for most of his childhood and adolescence, potentially contributing to the psychiatric comorbidities including depression, adjustment and eating disorder. The advances in medical care, including the expansion of treatment options due to interdisciplinary and multi-professional follow-up, have led to several reported cases of patients with LCCS11 with survival beyond infancy [4,7,9,10], and now even beyond adolescence. Therefore, the authors of this publication recommend that the stigmatizing term “lethal” in LCCS11 be reconsidered. Referring to the disease as FADS and specifying the detected GLDN variants could also be an option to avoid an additional psychological burden on affected patients.

## 4. Conclusions

LCCS11, associated with mutations of the gliomedin gene, which was formerly presumed to be lethal perinatally, can present as a chronic disease with aggravation of symptoms beyond childhood. For those patients, as well as patients with other neuromuscular diseases, diaphragmatic hypomotility can be a cause of increasing respiratory insufficiency in adolescence.

To establish a timely diagnosis for neuromuscular diseases, early genetic analyses, including exome sequencing, as well as regular follow-ups for new symptoms to adjust interdisciplinary treatment accordingly, are indispensable. Medical knowledge and the diagnostic possibilities have improved tremendously over the time span of our patient and are still getting better. “Unsolved” cases of neuromuscular disease should be invited for routine follow-up in specialized care centers with multi-disciplinary expertise; in our case, expertise in neuropediatrics and pediatric pulmonology as well as genetics. Multi-professional follow-up should be offered early in life to help to improve all dimensions of participation.

## Figures and Tables

**Figure 1 children-09-00797-f001:**
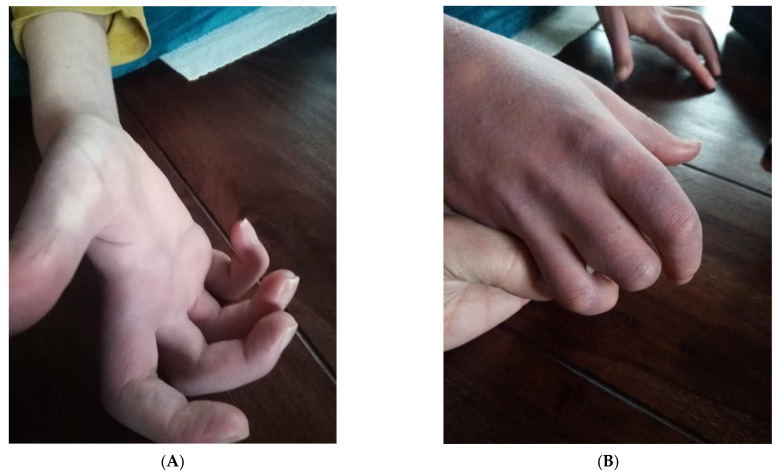
(**A**) The patient’s right hand showing contractures as well as palmar erythema, likely due to loss of autonomic function; (**B**) erythema of the patient’s right hand in contrast to his mother’s skin tone.

**Figure 2 children-09-00797-f002:**
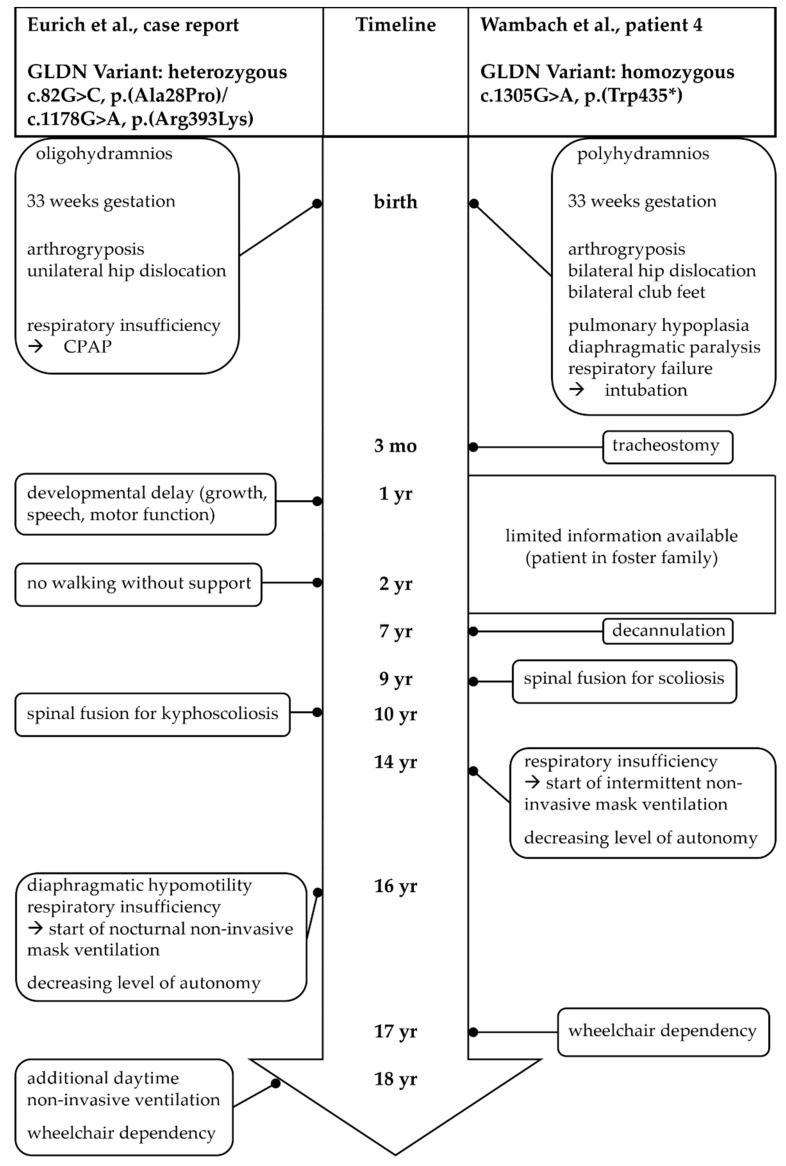
Comparison between LCCS11 patients with progressive clinical symptoms during adolescence; our patient and a patient reported by Wambach et al. [4].

## Data Availability

The data presented in this case report are available within the article.

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
