# Peer review of "Progressive Respiratory Insufficiency in a Teenager with Diaphragmatic Hypomotility Due to a Novel Combination of Gliomedin Gene Variants"

_children, 2022, doi:10.3390/children9060797_

Round 1

Reviewer 1 Report

The Case Report "Progressive Respiratory Insufficiency in a Teenager with Diaphragmatic Hypomotility Due to a Novel Combination of Gliomedin Gene Variants" by Benjamin Eurich et all is a solid description of a very rare pathology. 

There are only some small details that can improve the paper.

Page 2, line 54. You should point up the particularity of this this case discovered in such advanced stage and quite late in the adolescence. 

Page 3, line 92. Nocturnal home NIV commenced at home. Can you add some results with data from BiPAP card?

Page 6, line 242. Excellent suggestion to adjust the "lethal" therm. You need to mention again the importance of interdisciplinary and multiprofessional follow-up. 

Author Response

Thank you for your comments and suggestions.

Point 1: Page 2, line 54. You should point up the particularity of this this case discovered in such advanced stage and quite late in the adolescence.

Response 1: We have added a phrase to emphasize the particularity of this case:

"The presentation in late adolescence with such severe, chronic, and progressing signs and symptoms was unusual and led to a detailed diagnostic process following admission to the intermediate care ward."

Point 2: Page 3, line 92. Nocturnal home NIV commenced at home. Can you add some results with data from BiPAP card?

Response 2: Nocturnal home NIV had been started already when the patient presented at our clinic. Unfortunately, we do not have data from his home ventilator. Adjustments of the ventilation settings and modalities (as mentioned, inter alia introduction of a mouth piece, a volume guarantee and increasing the PIP) were carried out in a clinical setting (which is usually the case in our pediatric clinic, data from home ventilators are not routinely retrieved).

Point 3: Page 6, line 242. Excellent suggestion to adjust the "lethal" term. You need to mention again the importance of interdisciplinary and multiprofessional follow-up.

Response 3: We have amended the paragraph accordingly:

"The advances in medical care, including the expansion of treatment options due to interdisciplinary and multiprofessional follow-up, have led to several reported cases of patients with LCCS11 with survival beyond infancy [4,7,9,10], and now even beyond adolescence. Therefore, the authors of this publication recommend that the stigmatizing term “lethal” in LCCS11 be reconsidered."

Reviewer 2 Report

Thank you for the possibility to review the manuscript titled: “Progressive Respiratory Insufficiency in a Teenager with 2 Diaphragmatic Hypomotility Due to a Novel Combination of 3 Gliomedin Gene Variants”. The case report is interesting and easy to read. There are only a few minor corrections/recommendations.

The third and fourth paragraph in the section “2.4. Medical Course” should begin with a blank space.

Please add more detailes regarding the pathophysiology of respiratory failure in this disease (particularly combination of chest wall deformities and respiratory and muscle weakness causing altered chest wall mechanics). Discuss possible mechanisms to support these patients. Particulalry non-invasive ventilaton, invasive ventilation (in later stages), diaphragmatic stimulation.

The study is interesting, and provides information about a rare case. Please take into account the recommendation in the spirit of improving the quality of the submission.

Author Response

Thank you for your comments and suggestions.

Point 1: The third and fourth paragraph in the section “2.4. Medical Course” should begin with a blank space.

Response 1: We have amended the section accordingly.

Point 2: Please add more details regarding the pathophysiology of respiratory failure in this disease (particularly combination of chest wall deformities and respiratory and muscle weakness causing altered chest wall mechanics). Discuss possible mechanisms to support these patients. Particularly non-invasive ventilaton, invasive ventilation (in later stages), diaphragmatic stimulation.

Response 2: We have added a section discussing the pathophysiology as well as possible mechanisms to support these patients.

We have also inserted a phrase mentioning that our patient has received a cough assist device in the meantime.

"The exact pathophysiology of respiratory insufficiency in this disease has not been assessed formally, but reduced muscle stimulation due to axonopathy, leading to muscular atrophy and weakness, inclusive of intercostal muscles and the diaphragm, can be assumed a possible mechanistic explanation. Resulting diaphragmatic hypomotility may initially present as hypoventilation solely at night since diaphragmatic function is crucial for efficient ventilation during sleep [16]. Therefore, assessment of diaphragmatic function via sonography (to evaluate diaphragmatic motility and thickness) and/or twitch mouth (noninvasive) or transdiaphragmatic (invasive) pressure following cervical magnetic stimulation of the phrenic nerves as well as screening for nocturnal hypoventilation and hypoxemia should be considered as part of regular follow-ups in addition to pulmonary function testing in patients with signs of respiratory insufficiency [17-19]. Hiatal hernia, as seen in this patient, could be secondary to muscle wasting caused by diaphragmatic hypomotility and therefore a warning sign. Additionally, chest wall deformities, induced by limited fetal movement and exacerbated by wasting of trunk muscles, can aggravate restricted chest wall expansion and thereby inefficient inspiration as well as forced expiration, which can lead to insufficient clearing of mucus resulting in respiratory tract infections and dyspnea. Since patients with neuromuscular disease, especially when accompanied by chest wall deformity, are at risk of respiratory exacerbation, respiratory function should be monitored frequently to allow for early therapy adjustment.

Treatment options for respiratory insufficiency in LCCS11 (and neuromuscular disease in general) comprise ventilation, cough assistance, physiotherapy, and potentially diaphragm pacing. In early stages of the disease, when nocturnal hypoventilation is predominant, non-invasive ventilation may be sufficient. Using BiPAP rather than CPAP ventilation can reduce the work of breathing. If nocturnal hypoxemia is detected, oxygen supplementation can improve sleep quality. In later stages, when ventilation is required during daytime and almost continuously due to progressive diaphragmatic hypomotility and impaired chest wall expansion, non-invasive ventilation may interfere with communication and food intake, especially when ventilation through nasal prongs is insufficient and a mask required. Tracheostomy and subsequent invasive ventilation can alleviate these issues and reduce anatomic dead space. Cough-assisting appliances such as mechanical insufflation/exsufflation devices as well as manual techniques help with forced expiration and promote mucus clearance and hence prevent respiratory tract infections and aggravated dyspnea. Additionally, physiotherapy, specifically respiratory muscle training using pressure threshold breathing devices, has been shown to be effective in neuromuscular disease [20]. Diaphragm pacing following electrode implantation may potentially have a beneficial effect in patients with neuromuscular disease; however, diaphragm pacing has so far not proven to be neither effective nor safe in studies mainly conducted in ALS patients [21]."

"He has received a mechanical insufflation/exsufflation (MI-E) device to assist coughing and therefore support mucus clearance and prevent respiratory tract infections."

16. Berger, K.I.; Rapoport, D.M.; Ayappa, I.; Goldring, R.M. Pathophysiology of Hypoventilation During Sleep. Sleep Med Clin 2014, 9(3),289–300. [CrossRef]

17. Kabitz, H.J., Walterspacher, S.; Mellies, U.; Criee, C.P.; Windisch, W. (2014). Recommendations for Respiratory Muscle Testing. Pneumologie 2014, 68(5), 307-314. [CrossRef] [PubMed]

18. Spiesshoefer, J.; Henke, C.; Herkenrath, S.D.; Randerath, W.; Brix, T.; Görlich, D.; Young, P.; Boentert, M. Noninvasive Prediction of Twitch Transdiaphragmatic Pressure: Insights from Spirometry, Diaphragm Ultrasound, and Phrenic Nerve Stimulation Studies. Respiration 2019, 98(4), 301-311. [CrossRef] [PubMed]

19. Santos, D.B.; Desmarais, G.; Falaize, L.; Ogna, A.; Cognet, S.; Louis, B.; Orlikowski, D.; Prigent, H.; Lofaso, F. Twitch Mouth Pressure for Detecting Respiratory Muscle Weakness in Suspicion of Neuromuscular Disorder. Neuromuscul Disord 2017 27(6), 518-525. [CrossRef] [PubMed]

20. Aslan, G.K.; Gurses, H.N.; Issever, H.; Kiyan, E. Effects of Respiratory Muscle Training on Pulmonary Functions in Patients with Slowly Progressive Neuromuscular Disease: A Randomized Controlled Trial. Clin Rehabil 2013, 28(6), 573–581. [CrossRef] [PubMed]

21. Woo, A.; Tchoe, H.J.; Shin, H.W., Shin, C.M., Lim, C.M. Assisted Breathing with a Diaphragm Pacing System: A Systematic Review. Yonsei Med J 2020, 61(12), 1024. [CrossRef] [PubMed]